# Effects of Osthole on Progesterone Secretion in Chicken Preovulatory Follicles Granulosa Cells

**DOI:** 10.3390/ani10112027

**Published:** 2020-11-04

**Authors:** Na Sun, Yutong Zhang, Yaxin Hou, Yanyan Yi, Jianhua Guo, Xiaozhong Zheng, Panpan Sun, Yaogui Sun, Ajab Khan, Hongquan Li

**Affiliations:** 1College of Veterinary Medicine, Shanxi Agricultural University, Jinzhong 030801, China; snzh060511@126.com (N.S.); zhangyutong5566@163.com (Y.Z.); houyaxinxt@163.com (Y.H.); yieryan2015@163.com (Y.Y.); dkyypb@163.com (Y.S.); drajab22@gmail.com (A.K.); 2Department of Veterinary Pathobiology, College of Veterinary Medicine & Biomedical Sciences, Texas A&M University, College Station, TX 77843, USA; jguo@cvm.tamu.edu; 3Centre for Inflammation Research, Queen’s Medical Research Institute, The University of Edinburgh, Edinburgh EH16 4TJ, UK; Xiaozhong.Zheng@ed.ac.uk; 4Laboratory Animal Center, Shanxi Agricultural University, Jinzhong 030801, China; sunpp0505@163.com

**Keywords:** osthole, progesterone, chicken preovulatory follicles granulosa cell

## Abstract

**Simple Summary:**

Progesterone produced by granulosa cells regulates the diverse reproductive events in poultry. Osthole is a natural compound extracted from Cnidium. In this study, we confirmed Osthole up-regulated the progesterone secretion though elevating the expression of key proteins in the process of progesterone synthesis. These results indicate Osthole could be used in the pre-peak phase and (or) the peak phase to maximize the output of egg production in laying hens. Moreover, it provided a new idea that natural compounds may be the target library to screen the potential drugs used in poultry to increase the egg quality and yield.

**Abstract:**

Osthole (Ost) is an active constituent of Cnidium monnieri (L.) Cusson which possesses anti-inflammatory and anti-oxidative properties. It also has estrogen-like activity and can stimulate corticosterone secretion. The present study was aimed to check the role of Ost on progesterone (P4) secretion in cultured granulosa cells obtained from hen preovulatory follicles. Different concentrations (5, 2.5, and 1.25 µg/mL) of Ost was added to granulosa cells for 6, 12, 18, and 24 h to investigate the level of progesterone secretions using enzyme linked immunosorbent assay (ELISA). The results showed that progesterone secretion was significantly increased in cells treated with Ost at 2.5 μg/mL. Also, qRT-PCR showed that mRNA expression of steroidogenic acute regulatory protein (StAR) was significantly up-regulated by Ost at 2.5 μg/mL concentration. Cytochrome P450 side-chain cleavage (P450scc) and 3β-hydroxysteroid dehydrogenase (3β-HSD) was significantly up-regulated by Ost. However, no significant differences were observed for the expression of proliferating cell nuclear antigen (PCNA). The protein expression of StAR, P450scc and 3β-HSD were significantly up-regulated by Ost treatment. The concentration of cyclic adenosine monophosphate (cAMP) and protein kinase A (PKA) in cell lysates showed no change with Ost treatment at 2.5 μg/mL by ELISA. An ROS kit showed non-significant difference in the level of reactive oxygen species (ROS). In conclusion, Ost treatment at a concentration of 2.5 μg/mL for 24 h had significantly up-regulated P4 secretion by elevating P450scc, 3β-HSD and StAR at both gene and protein level in granulosa cells obtained from hen preovulatory follicles.

## 1. Introduction

With the rapid development of the poultry and egg industry, people are now focusing on the production and quality of eggs. In order to increase egg production, amoxicillin, ciprofloxacin and other drugs were used to prevent avian epidemics and increase egg production which has caused drug residues in eggs [1]. Moreover, due to indiscriminate use and inappropriate higher doses, antimicrobial drugs result in an accumulation of harmful residues in edible tissues of poultry which are potential direct threats to humans in the form of toxicity [1]. Due to ban of antibiotics in poultry feed, the world is now in search of new methods to improve egg production and quality. 

Traditional Chinese medicines have been used for centuries to prevent and cure diseases. The main bioactive component Osthole is isolated from the seeds of Cnidium monnieri (L.) Cusson having many biological effects, such as anti-oxidant, anti-inflammatory and anti-cancer activities [2]. Furthermore, Osthole had an estrogen-like effect and promoted the corticosterone biosynthesis and secretion [3,4].

Progesterone (P4) plays a crucial part in ovulation which is regulated by luteinizing hormone in pre-ovulatory surge. Follicles in ovary are divided into four types, including small white follicles, large white follicles, small yellow follicles and pre-ovulatory follicles [5]. In hens, P4 is solely produced by the granulosa cells obtained from pre-ovulatory follicles [6,7]. Synthesis of progesterone is initiated by the transfer of cholesterol from the outer mitochondrial membrane to the inner one by steroid acute regulatory proteins (StAR) [8,9]. Cytochrome P450 cholesterol side chain cleavage (P450scc) converts cholesterol into pregnenolone which runs out of the mitochondria and enters into smooth endoplasmic reticulum and is thus converted into progesterone by 3β-hydroxysteroid dehydrogenase (3β-HSD) [10,11].

In addition to direct effects on the process of progesterone synthesis, there are many other reasons which can indirectly affect the synthesis of progesterone. The proliferation of granulosa cells is closely related to the secretion of P4 [12,13]. Proliferating cell nuclear antigen (PCNA) plays an essential role in initiating cell proliferation and is thus a good indicator of cell proliferation [14]. P4 is a steroid hormone and its synthesis is mainly regulated by gonadotropins (e.g., follicle stimulatine hormone and luteinizing hormone) through a classical cAMP/PKA pathway. Intracellular cyclic adenosine monophosphate (cAMP) is activated by gonadotropins which causes the activation of protein kinase A (PKA) and specific transcription factors [15,16], resulting in the transcription and expression of steroid hormone-related proteins such as StAR and P450scc. Moreover, it is widely reported that reactive oxygen species (ROSs) inhibit steroidal hormone synthesis by robbing electrons [17,18,19]. 

In the current study, we investigated whether Ost up-regulated the secretion of P4 and explored the molecular mechanism involved.

## 2. Materials and Methods 

### 2.1. Isolation and Culture of Granulosa Cells

All animal experiments were performed under the regulation and guidelines of Animal Ethical Committees of Shanxi Agricultural University (Taigu, China) (Ethical code: SXAU-EAW-2020C0201). The chicken pre-ovulatory follicle granulosa cells were isolated according to the method described by Gilbert et al. with minor modifications [20]. In brief, the ovarian tissue from 200-day-old Hylan brown layer chickens with a laying rate of more than 90% were collected and washed with phosphate buffer saline (PBS)three times. Follicles larger than 9 mm were put onto a glass dishes, and the granulosa and theca layer were separated rapidly. After washing three times with PBS, the granulosa layers were cut into pieces (less than 1 mm^3^), placed in an ampoule containing 12.5 mg/mL of collagenase type II and incubated for 5 min at 37°C. An equal volume of pre-cooled M199 medium (Thermo fisher, Suzhou, Jiangsu, China) was added to inactivate collagenase type II. The digested granulosa layer was filtered through a 200 µm mesh, centrifuged at 1000 rpm for 8 min. The harvested cells were washed twice with M199 medium and trypan blue staining was used to determine the cell viability. Granulosa cells were grown in M199 medium containing 10% fetal bovine serum (FBS) (Sijiqing, Hangzhou, Zhejiang, China) and 1% penicillin-streptomycin for 24 h at 37°C and 5% atmospheric CO_2_. The grown cells were then used for the subsequent experiments and analysis. In the following experiment, four layer chickens were used to obtain enough granulosa cells.

### 2.2. Identification of Granulosa Cells by Immunofluorescence 

First, 0.5 mL granulosa cells with a density of 8 × 10^5^/mL were seeded into each laser scanning confocal dishes with 10% FBS and 1% penicillin-streptomycin mixed in M199 and incubated for 24 h. To block the granulosa cells, 3% BSA was used for 20 min at 37 °C. Follicle-stimulating hormone receptor (FSHR) rabbit polyclonal antibody (Proteintech, Wuhan, Hubei, China; Catalog number: 22665-1-AP) was added as the primary antibody and incubated for 2 h at 37 °C. The medium was discarded and cells were washed with PBS (three times) and incubated with second antibody-Mouse Anti-rabbit IgG/FITC antibody (Proteintech, Wuhan, Hubei, China) for immunofluorescence staining. Nuclei were stained with DAPI (Beyotime, Shanghai, China). 

### 2.3. Cell Viability

Ost was purchased from National Institutes for Food and Drug Control (NIFDC), China with 99.6% purity. Granulosa cells (0.1 mL) with a density of 8 × 10^5^/mL in 10% FBS and 1% penicillin-streptomycin mixed in M199 were seeded into 96-well-plate and incubated for 24 h. Ost was dissolved in 1% dimethyl sulfoxide (DMSO), and diluted to various concentrations (80, 40, 20, 10, 5, 2.5 and 1.25 μg/mL) with M199 containing 1% insulin-transferrin-Se (ITS; Sigma-Aldrich, St. Louis, MO, USA) and 1% penicillin-streptomycin, added into 96-well-plate and incubated for 48 h. Cell viability was performed by quantitative colorimetric assay with 3-(4,5-Dimethylthiazol-2-yl)-2,5-diphenyltetrazolium bromide (MTT) method. Briefly, the medium was discarded, 25 μL of MTT solution was added to each well and incubated for 4 h at 37 °C. The MTT was discarded and 150 μL of DMSO was added to each well for 30 min. With the help of microplate reader, the absorbance was measured at 490 nm and cell viability was calculated according to the absorbance value.

### 2.4. P4, cAMP and PKA assay

The P4, cAMP and PKA levels were determined by ELISA. 2 mL granulosa cells were seeded into a 6-well-plate at a density of 1 × 10^6^/mL in M199 with 10% FBS and 1% penicillin-streptomycin. After 24 h of incubation, 5, 2.5 and 1.25 μg/mL of Ost were added into the 6-well-plate. For P4, plates were respectively incubated for 6, 12, 18 and 24 h and for cAMP and PKA assay plates were incubated for 24 h. P4 (in the cell culture supernatant), cAMP and PKA (in cell lysates) were measured using chicken P4, cAMP and PKA ELISA kit (Bluegene, Shanghai, China), respectively according to the manufacturer’s instructions. 8-Bromo-cAMP (Topscience, Shanghai, China) was used as a positive control. 

### 2.5. Extraction of RNA and Quantitative Real-Time PCR

Cell samples were prepared as described in a section of cAMP ELISA. From the granulosa cells total RNA was extracted and reverse transcribed to cDNA using RNAiso Plus (Takara, Dalian, Liaoning, China) and PrimeScript^TM^ RT reagent Kit with gDNA Eraser (Takara, Dalian, Liaoning, China), respectively. The CDS regions of chicken origin PCNA, StAR, P450scc, 3β-HSD and β-actin were searched from the NCBI database. The respective primers were designed with Prime Premier 5 and synthesized by BGI (Beijing, China), as shown in Table 1. Then, qRT-PCR was performed using SYBR Green qPCR Master Mix (Bimake, Shanghai, China) by ABI 7500 real-time PCR machine. The relative quantities of StAR, P450scc, 3β-HSD and PCNA gene were calculated using 2^−△△Ct^ method, with β-actin used as a house-keeping gene. 

### 2.6. Western Blot

Cell samples were prepared as described in a section of cAMP ELISA. Protein concentrations of cell lysates were evaluated using a bicinchoninic acid (BCA)protein concentration detection kit (Beyotime, Shanghai, China). Next, 50 μg proteins were loaded and separated using 10% sodium dodecyl sulfate-polyacrylamide gel (SDS-PAGE) and transferred to a membrane of polyvinylidene difluoride (PVDF, 0.22 μm). The membrane was then blocked with Tris-buffered Tween 20 (TBST) with 5% non-fat dry milk. Western blot analysis was used to determine protein level of StAR (diluted 1:1000 with TBST, 4 °C overnight; Bioss, Beijing, China; Catalog number: A00051-1), P450scc (diluted 1:1000 with TBST, 4°C overnight; Bioss, Beijing, China; Catalog number: PB0983), 3β-HSD (diluted 1:1000 with TBST, 4°C overnight; Bioss, Beijing, China; Catalog number: A02856-2) and β-actin (diluted 1:4000 with TBST, room temperature 2h; Proteintech, Wuhan, Hubei, China; Catalog number: 66009-1-Ig). The membrane was three times washed with TBST and probed with HRP-conjugated secondary antibody. At the end, the proteins of interest were detected using an enhanced chemiluminescence system (Boster, Wuhan, Hubei, China). Image J software was used to analyze gray scales of Western blot images.

### 2.7. ROS Assay 

The granulosa cells (0.1 mL) were seeded into a 96-well-plate with a density of 8 × 10^5^/mL in 10% FBS and 1% penicillin-streptomycin mixed in M199 and incubated for 24 h. Next, 5, 2.5 and 1.25 μg/mL of Ost were added and incubated for another 24 h. Normal control and Rosup positive control were set up simultaneously. The ROS level in the cell lysates were detected by ROS Assay Kit (Beyotime, Shanghai, China) following manufacturer’s instructions. Briefly, the probe of DCFH-DA was diluted at 1:1000 with M199 and incubated for 20 min at 37 °C. After washing with M199 three times, cell fluorescence intensity was measured with the help of microplate reader at 488 nm and 525 nm excitation wavelength emission wavelength, respectively. 

### 2.8. Statistical Analyses 

Three experimental replicates were displayed and data are expressed as means ± SEM. One-way ANOVA was used to perform the statistical analysis followed by Bonferroni’s Multiple Comparison Test implemented in Graphad Prism 5 software (GraphPad Software, San Diego, CA, USA). *, * *, *** mean *p* < 0.05, *p* < 0.01 and *p* < 0.001, respectively.

## 3. Results

### 3.1. Morphology of Granulosa Cells

We observed morphology of granulosa cells at 24, 48 and 72 h after seeding the cells in the plates (Figure 1). The granulosa cells adhered to the surface of the plates and were able to grow within 24 h. The cells were irregular in shape and the pseudopods were connected to each other to aggregate and grow. Post 48 and 72 h of incubation, the granulosa cells were proliferated approximately 70% and 100%, respectively. The granulosa cells identity was confirmed by FSHR staining. The representative images of the fluorescence staining of FSHR in cells were shown in Figure 2. Strong staining was observed indicating the high expression of FSHR in the isolated granulosa cells.

### 3.2. Effect of Ost on Viability of Granulosa Cells 

Viability of granulosa cells were analyzed by MTT method and the results are shown in Figure 3. Compared with the control group, the cell viability was significantly decreased in groups treated with Ost at 80, 40, 20 and 10 μg/mL. Therefore, low concentrations of Ost (5, 2.5 and 1.25 μg/mL) were used in the follow up study in order to keep suitable cell viability.

### 3.3. Effect of Ost on P4 Secretion in Granulosa Cells 

In the supernatant, progesterone was measured at different concentrations (5, 2.5 and 1.25 μg/mL) and were respectively measured at 6, 12, 18 and 24 h. A significant increase was observed in P4 secretion after 24 h of incubation with 2.5 µg/mL of Ost only (*p* < 0.05; Figure 4). Therefore, 24 h treatment was selected for the follow up study.

### 3.4. Effect of Ost on mRNA Expression of PCNA, StAR, P450scc and 3β-HSD in Granulosa Cells

The results of qRT-PCR showed that PCNA mRNA expression was not significantly affected in Ost-treated groups as compared to the control group (Figure 5). StAR mRNA expression was significantly increased in the group treated with Ost at 2.5 μg/mL (*p* < 0.01; Figure 6A). P450scc and 3β-HSD were significantly increased in all the three groups treated with Ost at 5, 2.5, 1.25 μg/mL (Figure 6B,C). These results show that StAR, P450scc and 3β-HSD mRNA expression were up-regulated in chicken granulosa cells after 24 h of 2.5μg/mL Ost treatment.

### 3.5. Effect of Ost on Protein Expression of StAR, P450scc and 3β-HSD in Granulosa Cells

The expression of StAR, P450scc and 3β-HSD proteins in granulosa cells were significantly increased in western blot analysis (Figure 7A,C). The results confirm that the expression of StAR, P450scc and 3β-HSD proteins were consistent with that of mRNA expression in chicken granulosa cells after 24 h of 2.5 μg/mL Ost treatment.

### 3.6. Effect of Ost on cAMP and PKA Secretion in Granulosa Cells 

There was no significant difference in both cAMP and PKA secretion in cell lysates after 24 h of Ost treatment (Figure 8). The positive control 8-Bromo-cAMP increased both cAMP and PKA contents (*p* < 0.01). 

### 3.7. Effect of Ost on ROS Expression in Granulosa Cells

After 24 h of treatment, no significant difference was seen in ROS level between control and Ost-treated groups (Figure 9). The result indicated that Ost does not up-regulate P4 by down-regulating ROS after 24 h of treatment.

## 4. Discussion

In poultry, P4 is a hormone produced by granulosa cells which is obtained from pre-ovulatory follicles and regulates diverse reproductive events [5,6,7]. The study on regulation of progesterone synthesis and secretion has both theoretical and practical importance for the pre-peak phase and maximum output of peak egg production in laying hens. At present, Ost has been reported that can negatively affect the growth of some cancer cells [21,22,23], osteoporosis [24,25], inflammation [26,27,28] and possess estrogen-like effects [3,4]. But there are few research reports on the effect of Ost on reproductive endocrinology. The main purpose of this study was to check the effect of Ost on the regulation of P4 secretion. We tested three doses (5, 2.5, 1.25 μg/mL) of Ost with four incubation time points (6, 12, 18, 24 h). The results show that progesterone in granulosa cells obtained from pre-ovulatory follicles can be specifically upregulated at 24 h of incubation with 2.5 μg/mL of Ost. 

Cell proliferation is one of the key factors for the increase of progesterone secretion. PCNA is a nuclear protein which is necessary for DNA synthesis in eukaryotic cells, synthesized and stored in the nucleus [14]. This study revealed no significant difference in PCNA expression and thus confirmed that P4 up-regulation by Ost does not occur through regulation of cell proliferation. Xiao et al. [13] investigated the effects of genistein on P4 secretion in chicken granulosa cells harvested from follicles in vitro follocles, and the results show that P4 secretion was significantly stimulated by genistein through upregulation of P450scc, 3β-HSD and StAR gene transcription in cultured granulosa cells but the PCNA protein expression was not affected. Therefore, in our study, we focused on key enzymes in regulating the synthesis and secretion of progesterone. 

It is well known that cholesterol conversion to P4 is catalyzed by three key enzymes. Firstly, cholesterol enters into mitochondria from the outer membrane with the help of StAR. Then, the side chain cleavage of cholesterol was removed by P450scc to form pregnenolone which is transported out of the mitochondria. Finally, pregnenolone is converted to P4 by 3β-HSD [29,30]. Kisspeptin-10 can significantly stimulate P4 secretion and up-regulate StAR, P450scc and 3β-HSD gene transcription, but can’t up-regulate protein expression of StAR, P450scc and 3β-HSD in granulosa cells obtained from preovulatory follicles [31]. In our study, 24 h treatment of 2.5 μg/mL Ost has significantly up-regulated P4 by elevating StAR, P450scc and 3β-HSD expressions both at mRNA and protein level. 

Steroid hormones synthesis is monitored by gonadotropins through a classical cAMP/PKA pathway. Activation of cAMP/PKA signaling promotes StAR, P450scc and 3β-HSD expressions [15,16]. Wu et al. reported that steroidogenesis is inhibited by T2 toxin through suppressing of cAMP-PKA pathways and StAR is the main target for T-2-toxin [32]. Our results show that the contents of cAMP and PKA in cell lysates were not significantly changed post incubation of 24 h with Ost at three different concentrations. These results suggested that cAMP/PKA is not the main target of Ost. 

The process of progesterone synthesis is accompanied by mitochondrial electron transportation and oxidative phosphorylation. The mitochondrial membrane potential, H+ potential between the inner membrane and matrix played crucial roles in steroid hormones synthesis [33,34]. In the process of biological oxidation, most of the hydrogen removed by each substance is accepted by coenzymes. The hydrogen from these coenzymes passes through a series of electron transportations on the inner membrane of mitochondria to oxygen molecules to form water. ROSs are mainly produced by mitochondria during electron transport [35]. There have been several reports showed that ROSs can inhibit steroid hormone synthesis [17,18,19]. In this study, no significant difference was found at the level of ROS between treated and non-treated groups indicating that 2.5 μg/mL Ost did not play a role in ROS regulation. 

## 5. Conclusions

This was a base line study which reported that Ost significantly regulated P4 secretion through up-regulation of P450scc, 3β-HSD and StAR gene and protein expression. Cell proliferation, cAMP/PKA and ROS were not the main target of Ost to regulate progesterone secretion. In the future, we will further elucidate the target of Ost from hormone regulation. These results lay a theoretical foundation that Osthole may be developed as extending the peak period of laying eggs and increasing the egg yield of laying hens.

## Figures and Tables

**Figure 1 animals-10-02027-f001:**
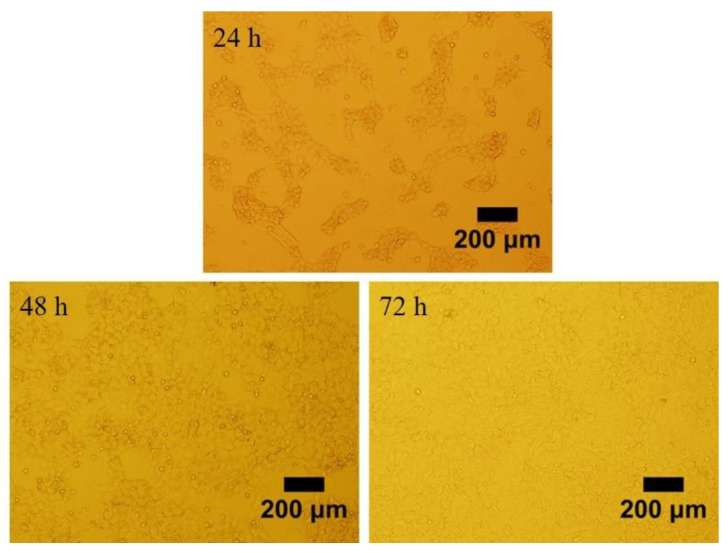
The preovulatory follicles of Hen’s granulosa cells cultured for 24, 48 and 72 h of post isolation. Scale bar, 200 μm.

**Figure 2 animals-10-02027-f002:**
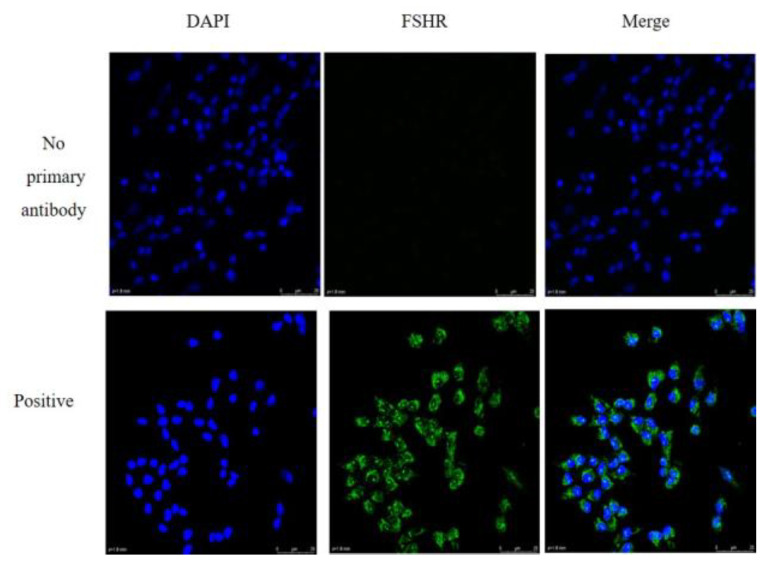
Fluorescence immunostaining for follicle-stimulating hormone receptor (FSHR) in granulosa cells. 4′,6-diamidino-2-phenylindole (DAPI )staining for nucleus (blue); immunoblotting for anti-FSHR cytoplasm (green); PBS for no primary antibody. Scale bar, 20 μm.

**Figure 3 animals-10-02027-f003:**
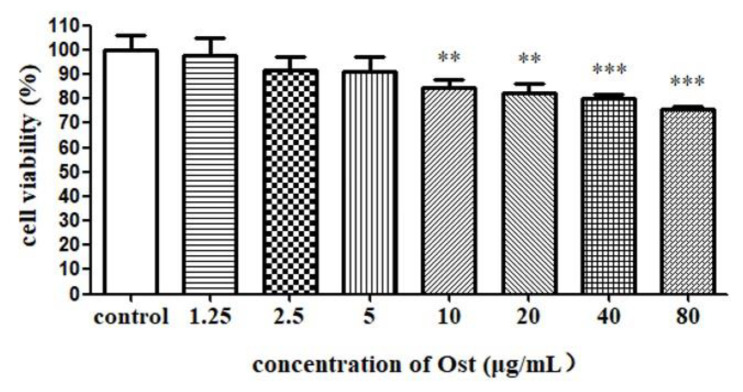
The cell viability of granulosa cells after treatment with Ost for 48 h. MTT was used to determine the cell viability. Data were expressed as Mean ± SEM. ** *p* < 0.01, *** *p* < 0.001.

**Figure 4 animals-10-02027-f004:**
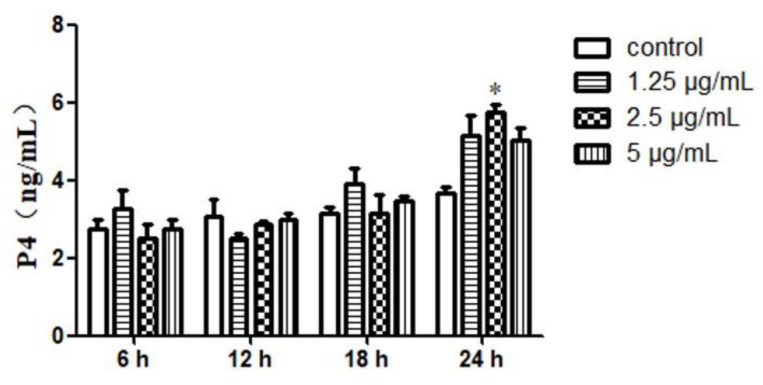
The level of P4 secretion in cultured granulosa cells treated with Ost at various time points. ELISA was used to determine the content of P4. Data were expressed as Mean ± SEM. * *p* < 0.05.

**Figure 5 animals-10-02027-f005:**
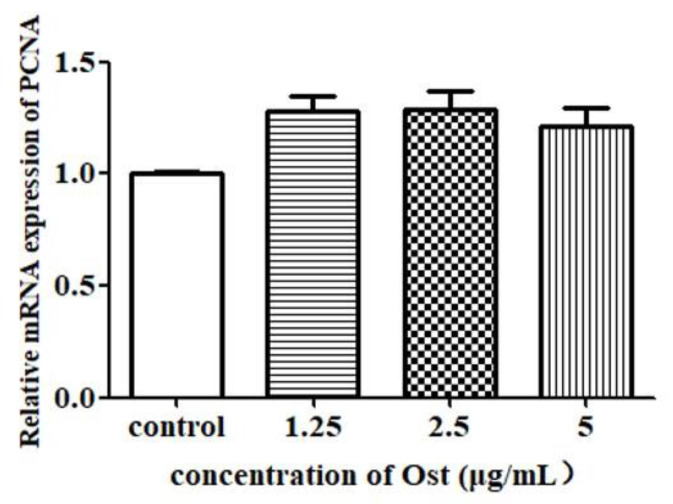
Effect of Ost on PCNA mRNA expression in granulosa cells. qRT-PCR was performed and 2^−△△Ct^ method was used for analysis. Data were expressed as Mean ± SEM.

**Figure 6 animals-10-02027-f006:**
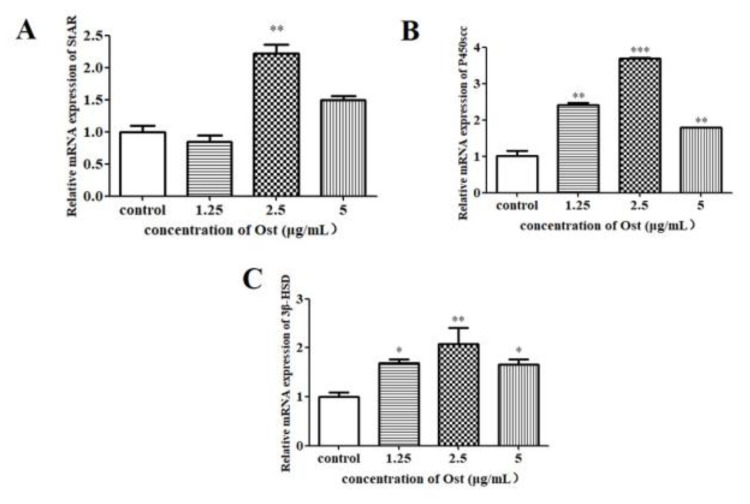
The mRNA expression of StAR (**A**), P450scc (**B**) and 3β-HSD (**C**) in granulosa cells treated with Ost for 24 h. qRT-PCR was performed and 2^−△△Ct^ method was used for analysis. Data were expressed as Mean ± SEM. * *p* < 0.05, ** *p* < 0.01, *** *p* < 0.001.

**Figure 7 animals-10-02027-f007:**
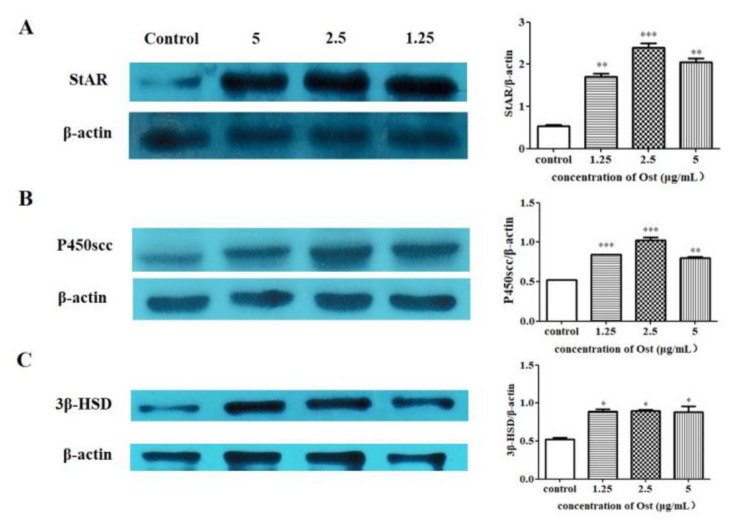
Western blot analysis of StAR (**A**), P450scc (**B**), and 3β-HSD (**C**) protein expression in granulosa cells after treatment with Ost (5, 2.5, 1.25 μg/mL) for 24 h. Image J software was used to analyze gray scales of Western blot images. Data were expressed as Mean ± SEM. * *p* < 0.05, ** *p* < 0.01, *** *p* < 0.001.

**Figure 8 animals-10-02027-f008:**
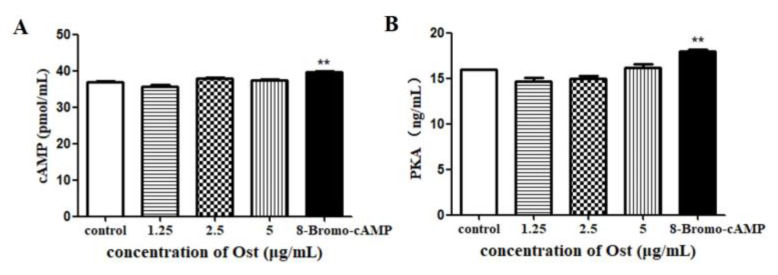
The level of cAMP(**A**) and PKA(**B**) in granulosa cells lysates were determined by ELISA. 8-Bromo-cAMP (Topscience, Shanghai, China) was used as a positive control. Data were expressed as Mean ± SEM. ** *p* < 0.01.

**Figure 9 animals-10-02027-f009:**
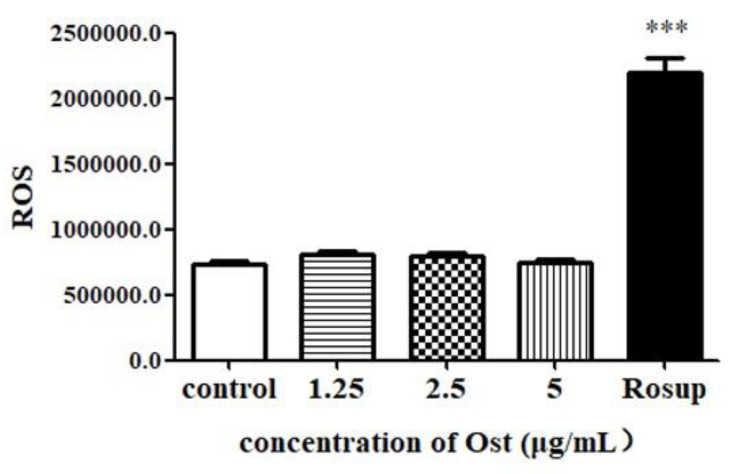
The production of reactive oxygen species (ROS) in granulosa cells post 24 h of treatment with Ost with Rousp was set as a positive control for ROS test. Data were expressed as Mean ± SEM. *** *p* < 0.001.

**Table 1 animals-10-02027-t001:** The primer sequences used in the present study.

Gene	cDNA Reference	Primer Sequences (5′→3′)	Annealing Temperature
β-Actin StAR P450scc 3β-HSD PCNA	NM_205518 NM_204686 NM_001001756 D43762 NM_204170.2	F: ATGAAGCCCAGAGCAAAAGA R: GGGGTGTTGAAGGTCTCAAA F: AGGGTTGGGAAGGACACTCT R: ATACATGTGGGGCCGTTCTC F: TCCGCTTTGCCTTGGAGTCTGTG R: ATGAGGGTGACGGCGTCGATGAA F: GCTTTGCCTTGGAGTCTGTG R: TCGGTGCTCTTGCGTTGC F: ATGGGCGTCAACCTAAACAG R: ATTCCAAGCTGCTCCACATC	60 °C 60 °C 60 °C 60 °C 60 °C

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
