# Peer review of "Effects of Osthole on Progesterone Secretion in Chicken Preovulatory Follicles Granulosa Cells"

_animals, 2020, doi:10.3390/ani10112027_

Round 1

Reviewer 1 Report

Line 68-69: “The proliferation of granulosa cells is closely related to the secretion of P4.” In which way? Typically P4 producing GC do not proliferate. Please explain in more detail and add reference.

Line 247: “Cell proliferation is one of the key factors for the increase of progesterone secretion.” That is not backed by any studies I am familiar with. Growth factors, like EGF and TGF, on the other hand are well known proliferative factors in GC. Please check and provide additional references on chicken GC proliferation factors.

Additional comments on incubation of GC in Ost: the production of P4 in GC is stimulated by LH binding to its receptor. It would have been interesting to see what the effect of Ost would have been on LH stimulated GC instead or in addition to unstimulated control.

Author Response

Dear Reviewer,

We highly appreciate your work on the manuscript with reference no. 975944. We have revised the manuscript according to your comments and uploaded the revised file. All the comments have been carefully considered and response to each point is present below.

  1. Line 68-69: “The proliferation of granulosa cells is closely related to the secretion of P4.” In which way? Typically P4 producing GC do not proliferate. Please explain in more detail and add reference.

Response: The granulosa cells from preovulatory follicles were proliferated in vitro culture system in our study. Actually, we could not explain that how it was happened in this study. But we did the growth curve analysis of granulosa cells in the pre-experiment of cell culture and the results showed that the proliferation was started from 48 h. Some other researchers have also confirmed it. For example, Heparin-binding EGF-like growth factor increased the cell number of differentiated granulosa cells from preovulatory follicles both using CellTiter 96 AQueous One solution cell proliferation assay kit and microscope (Wang et al, 2007). Granulosa cells proliferation was also used to assess the effect of some factors on P4 production (Li et al, 2019). Similar results were showed in Xiao YQ “Genistein significantly stimulating P4 secretion through the upregulation of ERβ, P450scc, 3β-HSD, and StAR gene transcription in cultured granulosa cells. And genistein did not affect the viability of granulosa cells, nor was the PCNA protein changed (Xiao et al, 2019)”. 

References

  1. Wang Y, Li J, Ying Wang C, Yan Kwok AH, Leung FC. Epidermal growth factor (EGF) receptor ligands in the chicken ovary: I. Evidence for heparin-binding EGF-like growth factor (HB-EGF) as a potential oocyte-derived signal to control granulosa cell proliferation and HB-EGF and kit ligand expression. Endocrinology. 2007, 148(7):3426-40.
  2. Li J, Luo W, Huang T, Gong Y. Growth differentiation factor 9 promotes follicle-stimulating hormone-induced progesterone production in chicken follicular granulosa cells. Gen Comp Endocrinol. 2019, 15;276:69-76.
  3. Xiao YQ, Shao D, Tong HB, Shi SR. Genistein increases progesterone secretion by elevating related enzymes in chicken granulosa cells. Poult Sci. 2019, 98(4):1911-1917.

  1. Line 247: “Cell proliferation is one of the key factors for the increase of progesterone secretion.” That is not backed by any studies I am familiar with. Growth factors, like EGF and TGF, on the other hand are well known proliferative factors in GC. Please check and provide additional references on chicken GC proliferation factors.

Response: P4 is produced by granulosa cells obtained from pre-ovulatory follicles in hens, so the number of granulosa cells will affect P4 contents. Because there was no effect of Ost on GC proliferation, so we discussed little about the proliferation factors.  

  1. Additional comments on incubation of GC in Ost: the production of P4 in GC is stimulated by LH binding to its receptor. It would have been interesting to see what the effect of Ost would have been on LH stimulated GC instead or in addition to unstimulated control.

Response: Thanks for your good ideas, and we will design experiments to assess the effect of Ost on LH or the combined effects of Ost and LH in the following studies. 

We all hope that our revised manuscript will meet with your approval.

Kind regards,

Na Sun

Reviewer 2 Report

In the article titled "Effects of osthole on progesterone secretion in chicken preovulatory follicles granulosa cells", the authors present results supporting a role for osthole (Ost) in progesterone secretion in chicken preovulatory follicles. The increase in progesterone secretion occurs through Ost-induced the upregulation of enzymes involved in progesterone synthesis and not through the increased proliferation of follicular granulosa cells. The study presents some interesting data and would have greatly benefited from the effect of Ost on egg quality.

I did not find the information regarding biological repeats used for each experiment.

Minor comments:

The article needs minor language corrections:

Line 23: it provided a new idea...

Line 37: The concentration of...

Lines 47-49, 68-69: Missing reference

Line 78: without 'hidden'

Lines 191-192: Sentence does not make sense, please modify it.

Line 192: A significant increase

Line 247: needs reference

Line 249: PCNA produced P4 levels???

Reference 27: Please cross-check this reference, it is not the correct one.

 Line 271: significantly

Author Response

Dear Reviewer,

We highly appreciate your work on the manuscript with reference no. 975944. We have revised the manuscript according to your comments and uploaded the revised file. All the comments have been carefully considered and response to each point is present below.

  1. I did not find the information regarding biological repeats used for each experiment.

Response: Thanks for your suggestions. Three experimental replicates were displayed in each assay and we have added the information in the section of statistical analyses.  

  1. Minor comments:

The article needs minor language corrections:

Line 23: it provided a new idea...

Line 37: The concentration of...

Line 78: without 'hidden'

Line 192: A significant increase

Line 271: significantly

Response: Thanks for your corrections and we have improved the language of our manuscript and have revised the minor comments in the new submission.

Lines 47-49, 68-69: Missing reference

Response: We have added relevant references in the revised manuscript.

Lines 191-192: Sentence does not make sense, please modify it.

Response: The sentence has changed to “In the supernatant, progesterone was measured at different concentration (5, 2.5 and 1.25 μg/mL) for 6, 12, 18 and 24 h.”

Line 247: needs reference

Response: The sentence was just the result of our study.

Line 249: PCNA produced P4 levels???

Reference 27: Please cross-check this reference, it is not the correct one.

Response: Thanks for your suggestion and we deleted the sentence and ref.27.

We all hope that our revised manuscript will meet with your approval.

Kind regards,

Na Sun
